# Procalcitonin as a Predictive Tool for Death and ICU Admission among Febrile Neutropenic Patients Visiting the Emergency Department

**DOI:** 10.3390/medicina58080985

**Published:** 2022-07-23

**Authors:** Christopher J. Coyne, Edward M. Castillo, Rebecca A. Shatsky, Theodore C. Chan

**Affiliations:** 1Department of Emergency Medicine, University of California San Diego, San Diego, CA 92103, USA; emcastillo@health.ucsd.edu (E.M.C.); tcchan@health.ucsd.edu (T.C.C.); 2Department of Medicine, Division of Hematology/Oncology, University of California San Diego, San Diego, CA 92037, USA; rshatsky@health.ucsd.edu

**Keywords:** procalcitonin, febrile neutropenia, neutropenic fever, emergency department, oncologic emergencies, cancer, neutropenia, sepsis, biomarker

## Abstract

*Background and Objectives*: Risk stratification tools for febrile neutropenia exist but are infrequently utilized by emergency physicians. Procalcitonin may provide emergency physicians with a more objective tool to identify patients at risk of decompensation. *Materials and Methods*: We conducted a retrospective cohort study evaluating the use of procalcitonin in cases of febrile neutropenia among adult patients presenting to the Emergency Department compared to a non-neutropenic, febrile control group. Our primary outcome measure was in-hospital mortality with a secondary outcome of ICU admission. *Results*: Among febrile neutropenic patients, a positive initial procalcitonin value was associated with significantly increased odds of inpatient mortality after adjusting for age, sex, race, and ethnicity (AOR 9.912, *p* < 0.001), which was similar, though greater than, our non-neutropenic cohort (AOR 2.18, *p* < 0.001). All febrile neutropenic patients with a positive procalcitonin were admitted to the ICU. Procalcitonin had a higher sensitivity and negative predictive value (NPV) in regard to mortality and ICU admission for our neutropenic group versus our non-neutropenic control. *Conclusions*: Procalcitonin appears to be a valuable tool when attempting to risk stratify patients with febrile neutropenia presenting to the emergency department. Procalcitonin performed better in the prediction of death and ICU admission among patients with febrile neutropenia than a similar febrile, non-neutropenic control group.

## 1. Introduction

Febrile Neutropenia is a feared complication of chemotherapy and is associated with high morbidity and mortality among patients with cancer. Over the past half-century, medical researchers have dedicated significant effort towards elucidating which patients with febrile neutropenia are at the greatest risk of decompensation [1,2,3]. Often, patients with febrile neutropenia fail to exhibit the classical exam findings when suffering from potentially life-threatening bacterial illnesses due to their compromised immune systems [4]. Therefore, these patients have historically been almost exclusively admitted to the hospital after presenting to the emergency department (ED) [5]. More recently, medical researchers and clinicians have recognized that there is likely a cohort of febrile neutropenic patients that do not require this highly aggressive/conservative strategy [6,7,8,9]. These lower-risk patients may actually experience harm from being admitted to the hospital, including exposure to nosocomial infections and emotional distress, as well as expulsion from therapeutic cancer clinical trials, while concurrently subjecting them to invasive studies and the associated costs of a lengthy hospital stay.

Clinician scientists have been attempting to risk stratify patients with febrile neutropenia for several decades. In the 1980s, Talcot et al. created a risk stratification score that allowed clinicians to identify patients at a high risk of clinical deterioration when experiencing febrile neutropenia [10,11]. Over the past two decades, two additional scores have been created to more accurately risk stratify patients with febrile neutropenia: The Multinational Association for Supportive Care in Cancer (MASCC) Risk Index and the Clinical Index of Stable Febrile Neutropenia (CISNE) score [2,9]. In the most recent update on the management of febrile neutropenia, the National Comprehensive Cancer Network (NCCN) guidelines suggest that these scores can be applied under the correct circumstances to aid clinicians in risk stratification efforts [12]. Patients that are found to be low risk may be safely discharged with oral antibiotics and close follow-up. Unfortunately, most emergency physicians continue to admit all patients with febrile neutropenia, even those considered low risk, despite the potential iatrogenic complications.

Several biomarkers have been proposed as potentially useful tools to help differentiate febrile neutropenic patients at a high risk of clinical deterioration, including presepsin, IL-6, IL-8, adrenomedullin, and CRP, among many others [13,14,15,16,17,18,19,20,21,22,23]. All of these have their potential benefits; however, they all appear to fall short, either by lacking sensitivity/specificity or by lacking availability in the timeframe needed to make clinical decisions in the ED. In recent years, however, procalcitonin has become more widely available in healthcare systems. Importantly, in-hospital laboratories are generally able to process procalcitonin within the timeframe necessary to make important decisions regarding antibiotic utilization and the need for admission. Previous studies have suggested that procalcitonin may be useful in risk stratifying patients with febrile neutropenia [14,15,16,23,24]. However, many of these investigations represent a single site and lack statistical power. Additionally, it remains unclear how well procalcitonin performs among immunocompromised patients versus a similar immunocompetent control group. Our objective is to compare the accuracy of procalcitonin in the prediction of morbidity and mortality among febrile neutropenic patients compared to a similar group of febrile non-neutropenic patients. 

## 2. Materials and Methods

### 2.1. Study Design and Selection of Participants

We conducted a retrospective cohort study evaluating the use of procalcitonin in cases of febrile neutropenia compared to a non-neutropenic, febrile control group. We included all patients ≥18 years of age who presented to one of our study EDs, which included two academically affiliated EDs in Southern California, USA, between 1 January 2017 and 30 December 2021 who had a temperature of 100.4 °F (38.0 °C) or higher and also received a procalcitonin test. Patients were considered neutropenic if their absolute neutrophil count was <1000 cells/mm^3^. We chose this time period, because procalcitonin became more widely adopted in our study EDs at the beginning of 2017. and our dataset was complete up to the end of 2021. 

### 2.2. Methods of Measurement

Data were collected by study coauthor EC through an electronic medical records (EMR) system, as well as manual chart review when necessary. Procalcitonin values were considered positive if they were ≥0.25 ng/mL, which is the laboratory cutoff at our affiliated study hospitals. We defined neutropenia as <1000 cells/mm^3^ based on the previous literature [7]. Our primary outcome measure was in-hospital mortality, which was assessed through. Our secondary outcome was admission to the intensive care unit. We collected demographic variables for all patients including age, biological sex, self-reported race, and ethnicity. Strengthening the Reporting of Observational Studies in Epidemiology (STROBE) methods were utilized when conducting the reporting of this study [25]. We defined all variables a priori, and a Cohen’s kappa score was performed on 5% of patient charts to assess the inter-rater reliability between authors EC and CC using ICU admission as our variable of interest. This resulted in a kappa of 0.88, which represents substantial agreement. All discrepancies were discussed by all authors, and consensus was achieved. 

### 2.3. Data Analysis

We conducted a multivariable logistic regression analysis to determine whether an elevated initial procalcitonin level was associated with in-hospital mortality or admission to the ICU. We chose this analysis to allow for the adjustment of several potential confounding variables, including age, sex, race, and ethnicity. We conducted these analyses for the overall cohort and then separately for the neutropenic and non-neutropenic groups. We determined the significance at an alpha of 0.05. All statistical analyses were conducted by study author CC using IBM SPSS Version 26. This study was approved by our affiliate sites’ institutional review board. 

## 3. Results

We identified 4434 patients who met our inclusion criteria during our study period, 198 of whom were neutropenic. Our neutropenic group was similar to our control group with respect to age (median 59 in both) and race, with the neutropenic group being 47.5% White, 8.1% Black, and 13.1% Asian and the non-neutropenic control group being 46.8% White, 10.9% Black, and 8.3% Asian (Table 1). The neutropenic group had slightly more female patients (44% vs. 40.5%), while the control group had more Hispanic patients (29.7% vs. 24%). The neutropenic group experienced a higher percentage of ICU admissions (5.1% vs. 4.2%), as well as a higher percentage of deaths (7% vs. 3.7%).

Among the total cohort, a positive initial procalcitonin value was associated with increased odds of inpatient mortality after adjusting for age, sex, race, and ethnicity (AOR 2.37, *p* < 0.001) (Table 2). Similarly, a positive initial procalcitonin value was associated with an increased odds of ICU admission after similar adjustment (AOR 1.75, *p* = 0.001).

Among our control group, we identified similar outcomes. Specifically, a positive initial procalcitonin value was associated with increased odds of inpatient mortality after adjusting for age, sex, race, and ethnicity (AOR 2.18, *p* < 0.001). Similarly, a positive initial procalcitonin value was associated with increased odds of ICU admission after adjusting for the same factors (AOR 1.62, *p* = 0.003).

Among the patients with a neutropenic fever, we identified similar significant associations. A positive initial procalcitonin value was similarly associated with increased odds of inpatient mortality after adjusting for age, sex, race, and ethnicity (AOR 8.75, *p* = 0.39). This AOR indicates that a febrile, neutropenic patient with a positive procalcitonin may has nearly nine times the odds of inpatient mortality versus a patient with a negative procalcitonin. All patients with a positive procalcitonin were admitted to the ICU.

When comparing the procalcitonin test characteristics between our neutropenic and control groups for our primary outcome, we found that procalcitonin had a higher sensitivity and negative predictive value (NPV) in regard to mortality (Table 3). Specifically, among neutropenic patients, procalcitonin had a sensitivity of 92.9% and an NPV of 98.7% compared to the control group, where procalcitonin had a sensitivity of 74.4% and an NPV of 97.8%.

## 4. Discussion

Our data suggest that procalcitonin may be a useful test in the risk stratification of patients with febrile neutropenia. Although some have suggested that immunocompromised patients may experience an inconsistent rise in inflammatory biomarkers, our data suggest that procalcitonin elevation is quite uniform, with a negative predictive value of 98.7% in regard to mortality. Furthermore, the procalcitonin test characteristics for patients with febrile neutropenia appear to be better than those for the non-neutropenic control group in our study, suggesting that this biomarker may be particularly useful in this rare, though potentially fatal, condition.

In 2019, Baugh et al. published a study evaluating 348,868 ED visits for febrile neutropenia, utilizing the National Emergency Department Sample (NEDS) [5]. Among this group, 94% of patients were hospitalized, despite several studies indicating that approximately 20–30% of patients presenting to EDs with febrile neutropenia are likely appropriate for outpatient management. Why are all of these patients being admitted to the hospital when the national guidelines suggest that a substantial amount of low-risk patients should be discharged [12]? The authors of this article suggest that the currently available risk stratification tools are too cumbersome. Even in cases when the MASCC score is applied and a patient is found to be low risk, dozens of additional criterial need to be in order to safely discharge a patient. Additionally, the CISNE score is only applicable to patients with solid tumors, leaving out the large proportion of patients with hematological malignancies [7,8,9]. This is where biomarkers may play a critical role, either in addition to these clinical decision rules or as a primary risk stratification tool, similar to a D-Dimer test for pulmonary embolism [26,27,28].

Previous investigations into the use of procalcitonin as a risk stratification tool for patients with febrile neutropenia have been favorable [13,14,15]. Much of the literature, however, focuses on pediatric patients, with only few studies investigating the use of procalcitonin in an adult febrile neutropenic cohort [16,17,18,19]. Previous adult studies have investigated alternative biomarkers in febrile patients, such as C-Reactive Protein (CRP); however, these markers may lose their predictive ability among immunocompromised patients [14,15].

Among our cohort, we found that patients with febrile neutropenia had a higher ICU admission rate and a higher inpatient mortality than the non-neutropenic, febrile control group. This is to be expected, given the immunocompromised status of these patients, and these findings are consistent with national febrile neutropenia mortality, which is approximately 10% [29]. What is most interesting about our results is that procalcitonin not only appears to be noninferior as a predictive tool for mortality among febrile neutropenic patients compared to a non-neutropenic cohort but actually also appears to be more sensitive in the immunocompromised group. For in-hospital mortality among the febrile neutropenic group, procalcitonin had a sensitivity of 92.9% and a NPV of 98.7%, which outperformed the MASCC and CISNE scores based on the original validation studies [2,30]. The procalcitonin test characteristics for ICU admission among the neutropenic group were similarly impressive, with the sensitivity and NPV both being 100%.

There are several potential limitations to this study. This was a retrospective analysis and therefore limited in the data elements available in the electronic medical record. It is possible that we may have missed episodes of febrile neutropenia due to data capturing error. Given that both sites were NCCN Cancer Center-affiliated and were both in a specific area of the United States, this study may lack external validity. As an ED-focused study, we only utilized the initial procalcitonin value for use in our predictive models. It is possible that subsequent in-hospital procalcitonin values may have been elevated beyond our <0.25 cutoff, leading to a possible misclassification bias. This will be evaluated in future studies.

## 5. Conclusions

Procalcitonin appears to be a valuable tool when attempting to risk stratify patients with febrile neutropenia presenting to the emergency department. This study demonstrates that a procalcitonin cutoff of <0.25 ng/mL in patients with febrile neutropenia is associated with a NPV of 98.7% with respect to mortality. These results suggest that procalcitonin may play a role in helping emergency physicians to identify low-risk patients appropriate for safe discharge. This study further adds to the existing body of literature by validating the use of procalcitonin in febrile neutropenic patients presenting to the ED when compared to a similar, non-neutropenic control group. Future, prospective, multi-center studies are needed to critically evaluate the use of this biomarker. In conjunction with other risk stratification tools like the CISNE and MASCC scores, procalcitonin may allow emergency physicians to safely discharge a low-risk cohort of febrile neutropenic patients.

## Figures and Tables

**Table 1 medicina-58-00985-t001:** Demographics and Outcome Frequencies.

	ANC < 1000 cells/mm^3^ *N* = 198	ANC ≥ 1000 cells/mm^3^ *N* = 4236	Total Cohort *N* = 4434
Age	Mean 56.84, Median 59	Mean 57.41, Median 59	Mean 57.38 Median 59
Sex	*n* (%)	*n* (%)	*n* (%)
Female	88 (44)	1717 (40.5)	1805 (40.7)
Male	110 (66)	2519 (59.5)	2629 (59.3)
Race	*n* (%)	*n* (%)	*n* (%)
White	94 (47.5)	1981 (46.8)	2075 (46.8)
Black	16 (8.1)	462 (10.9)	478 (10.8)
Asian	26 (13.1)	353 (8.3)	379 (8.5)
Pacific Islander	1 (0.5)	13 (0.3)	14 (0.3)
Other	61 (30.8)	1427 (33.7)	1488 (33.6)
Ethnicity	*n* (%)	*n* (%)	*n* (%)
Non-Hispanic	149 (75.2)	2947 (69.6)	3096 (69.8)
Hispanic	49 (24.8)	1289 (30.4)	1338 (30.2)
Level of Care	*n* (%)	*n* (%)	*n* (%)
Med/Surg	6 (3)	154 (3.6)	160 (3.6)
Telemetry	149 (75.3)	3218 (76)	3367(75.9)
Stepdown	33 (16.6)	686 (16.2)	719 (16.2)
ICU	10 (5.1)	178 (4.2)	188 (4.3)
Initial Procalcitonin	*n* (%)	*n* (%)	*n* (%)
+(≥ 0.25 ng/mL)	123 (62.1)	2402 (56.7)	2525 (56.9)
−(< 0.25 ng/mL)	75 (37.9)	1834 (43.3)	1909 (43.1)
Mortality	*n* (%)	*n* (%)	*n* (%)
	14 (7)	156 (3.7)	170 (3.8)
Average ED Length of Stay (median)	623 min	614 min	614.5 min

Abbreviations: Absolute neutrophil count (ANC); Medical/Surgical Ward (med/surg); Intensive Care Unit (ICU).

**Table 2 medicina-58-00985-t002:** Odds of mortality and ICU admission for patients with a positive procalcitonin (≥0.25 ng/mL) compared to those with a negative procalcitonin.

	Mortality	ICU Admission
ANC < 1000	AOR 8.75 *p* = 0.39	NA, all + procalcitonin admitted to ICU
ANC ≥ 1000	AOR 2.18 *p* < 0.001	AOR 1.62 *p* = 0.003
Total Cohort	AOR 2.37 *p* < 0.001	AOR 1.75 *p* = 0.001

Abbreviations: Absolute Neutrophil Count (ANC); Adjusted Odds Ratio (AOR); Not Applicable (NA); Intensive Care Unit (ICU).

**Table 3 medicina-58-00985-t003:** Procalcitonin test characteristics for mortality and ICU admission.

	Mortality				ICU Admission			
	Sensitivity	Specificity	NPV	PPV	Sensitivity	Specificity	NPV	PPV
ANC < 1000	92.9%	40.2%	98.7%	10.6%	100%	39.9%	100%	8.1%
ANC ≥ 1000	74.4%	44.0%	97.8%	4.8%	68%	43.8%	96.9%	5%
Total Cohort	75.9%	43.8%	97.9%	5.1%	69.7%	43.6%	97.0%	5.2%

Abbreviations: Absolute Neutrophil Count (ANC); Negative Predictive Value (NPV); Positive Predictive Value (PPV).

## Data Availability

Not Applicable.

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
