# Peer review of "Procalcitonin as a Predictive Tool for Death and ICU Admission among Febrile Neutropenic Patients Visiting the Emergency Department"

_medicina, 2022, doi:10.3390/medicina58080985_

Round 1

Reviewer 1 Report

1. The research question is clearly stated. If risk stratification tools for febrile neutropenia already exist then why it is frequently utilized by emergency physicians.

2. Reason for choosing multivariable logistic regression analysis method is not stated.

3. Criteria for selecting study participants are not explained and justified.

4. Recruitment methods are not explicitly stated.

5. Methods are not outlined and examples are not given.

6. Data analysis and verification are not described, including by whom they were performed.

7. Methods for identifying/extrapolating themes and concepts from the data are not discussed.

8. Sufficient data are not presented to allow a reader to assess whether or not the interpretation is supported by the data.

9. Transferability of research findings to other settings is not discussed.

10. Any particular strengths and limitations of the research are not discussed.

11. In the conclusion, please states the main findings of the study and emphasize what the study adds to knowledge in the subject area.

Reviewer 2 Report

Work by Coyne and colleagues aims to determine the utility of procalcitonin as a predictive marker in febrile neutropenia. While this seems to be a subject of previous research (e. g. Haddad et al. Sci Rep 2018; Sakr et al., Infection 2008; Robinson et al., PLoS One 2011), Authors provide novel evidence that levels of procalcitonin, with a cut-off of .25, may be a strong predictor for clinical outcomes in febrile neutropenia. From this standpoint, proposed research is an important report for clinical community, and I have only some minor points that should be addressed prior to publication:

- in line 79 units for the temperature are not provided; also, I believe it is more common to report bodily temperature in Celsius degrees rather than Fahrenheit;

- in Table 1 in distrubution of sex among the whole cohort there is no numerical value for percents, only for n;

- in line 132 there is a double space;

- in line 149 PE abbreviation (I believe Pulmonary Embolism) is not described.

Round 2

Reviewer 1 Report

I read the revised manuscript. The authors had clealy responsed to my previous suggestions. I had no further comments.